# Targeting PI3K Signaling to Overcome Tumor Immunosuppression: Synergistic Strategies to Enhance Cancer Vaccine Efficacy

**DOI:** 10.3390/vaccines13030292

**Published:** 2025-03-10

**Authors:** Ran Cui, Zhongxiang Luo, Xialin Zhang, Xinlin Yu, Gang Yuan, Xingming Li, Fei Xie, Ou Jiang

**Affiliations:** 1Department of Oncology, The First People’s Hospital of Neijiang, Neijiang 641000, China; cr100492@hotmail.com (R.C.); lixingming120120@163.com (X.L.); xmahcu@163.com (F.X.); 2Department of Oncology, Southwest Medical University, Luzhou 646000, China; lzxxswmu2023@126.com (Z.L.); zxl499589252@163.com (X.Z.); 3Department of Oncology, Affiliated Hospital of Chengdu University, Chengdu 610000, China; yuxinlin1998@163.com; 4Department of Interventional & Vascular, Affiliated Traditional Chinese Medicine Hospital of Southwest Medical University, Luzhou 646000, China; gyuan@swmu.edu.cn

**Keywords:** PI3K, cancer, vaccine

## Abstract

Phosphoinositide 3-kinases (PI3Ks), members of the lipid kinase family, play a significant role in modulating immune cell functions, including activation, proliferation, and differentiation. Recent studies have identified the PI3K signaling pathway as a key regulator in tumor biology and the immune microenvironment. This pathway enhances the activity of regulatory T cells (Tregs) and myeloid-derived suppressor cells (MDSCs), contributing to an immunosuppressive tumor microenvironment that impairs the effectiveness of cancer vaccines and immunotherapies. The present study explores PI3K isoforms, particularly p110γ and p110δ, and their associated signaling pathways. The therapeutic potential of selective PI3K inhibitors and their capacity to act synergistically with immunization strategies are analyzed. Targeting the PI3K signaling pathway represents a promising approach to counteract tumor-induced immune suppression and improve the efficacy of immune checkpoint inhibitors and vaccines, ultimately leading to better clinical outcomes.

## 1. Introduction

Recent advancements in immunotherapy have highlighted the potential of cancer vaccines as critical tools for reprogramming the immune response against tumors. Personalized cancer vaccines, designed to align with the antigenic profiles of individual tumors, have demonstrated the ability to elicit robust and durable immune responses [1]. Furthermore, the combination of cancer vaccines with immune checkpoint inhibitors, such as PD-1/PD-L1 blockade, has shown efficacy in mitigating immunosuppressive mechanisms within the tumor microenvironment (TME), thereby enhancing vaccine effectiveness [2]. Tumors employ multiple strategies to evade immune detection and suppress immune function, including the establishment of an immunosuppressive TME, exploitation of tumor antigen heterogeneity, and activation of immune evasion pathways [3,4,5]. These challenges significantly limit the efficacy of immunotherapies, particularly cancer vaccines.

The PI3K signaling pathway has emerged as a promising therapeutic target due to its involvement in both tumor progression and immune suppression. This review aims to explore the role of the PI3K signaling system in tumor immunity and its influence on cancer vaccine efficacy. By analyzing the interaction between PI3K signaling and immunological responses, insights into how modulation of this pathway may enhance immunotherapeutic outcomes can be gained. This study will also examine ongoing clinical trials investigating the combination of PI3K inhibitors with cancer vaccines, along with the associated challenges and opportunities. Emphasis will be placed on the potential of PI3K-targeted therapies as a fundamental component in the development of next-generation cancer vaccines and combination immunotherapies.

The present study makes significant contributions to the field of PI3K pathway research through three primary innovations. Firstly, it provides a comprehensive elucidation of the PI3K signaling cascade, systematically integrating molecular mechanisms with clinical applications to establish a multidimensional framework for understanding its oncogenic potential and therapeutic targeting. Secondly, the work advances current knowledge through the meticulous integration of clinical trial data, offering critical insights into inhibitor efficacy while objectively delineating pharmacological challenges across diverse cancer subtypes. Thirdly, this research includes pioneering new perspectives by elucidating the pathway’s immunomodulatory functions and synergies with emerging immunotherapies, particularly highlighting combinatorial potential with tumor vaccine strategies.

The findings underscore several critical implications for translational oncology. The demonstrated heterogeneity of PI3K dysregulation mechanisms across malignancies necessitates personalized therapeutic approaches, emphasizing the biomarker-driven selection of isoform-specific inhibitors. This work elaborates on a strategic therapeutic paradigm shift through its evidence-based analysis of combination therapies, proposing that PI3K inhibition may potentiate immunotherapy by remodeling the tumor microenvironment. Future research directions emphasize three key axes: development of next-generation inhibitors with enhanced specificity and delivery systems, optimization of combination protocols through mechanistic studies, and implementation of AI-driven platforms for predictive biomarker discovery. These insights collectively establish a roadmap for overcoming current therapeutic limitations while advancing precision oncology frameworks.

## 2. The Relationship Between the PI3K Signaling Pathway and Tumor Immunity

To sustain the function and longevity of immunosuppressive cells, tumor cells activate the PI3K signaling pathway, including regulatory T cells (Tregs) and myeloid-derived suppressor cells (MDSCs), thereby preserving an immunosuppressive tumor microenvironment [6]. Within signal transduction processes, the PI3K/Akt/mTOR signaling pathway plays a critical role in regulating Tregs by enhancing the production of Foxp3, a key transcription factor necessary for Treg differentiation and suppressive function [7]. Activation of the PI3K pathway through Akt stabilizes Foxp3 and increases its transcriptional activity, reinforcing Treg-mediated immunosuppression. Additionally, Akt activation enhances glycolysis and lipid metabolism in Tregs, supplying the metabolic support essential for their survival and suppressive activity within the nutrient-deprived TME [8]. These metabolic adaptations enable Tregs to outcompete effector T cells, maintaining an immunosuppressive state [9]. Similarly, the PI3K pathway is integral to the recruitment, activation, and immunosuppressive function of MDSCs [10]. The PI3K/Akt/mTOR pathway modulates MDSC metabolism by promoting oxidative phosphorylation and fatty acid oxidation, both of which are crucial for MDSC activity in the TME [11]. These metabolic alterations facilitate the production of immunosuppressive molecules, including reactive oxygen species (ROS), arginase, interleukin-10 (IL-10), and transforming growth factor-β (TGF-β), which inhibit T cell activation and proliferation [12]. Moreover, PI3K-induced Akt activation enhances the nuclear localization of STAT3 [13], a transcription factor that promotes MDSC expansion and immunosuppressive function, creating a positive feedback loop that perpetuates immune evasion. These mechanisms highlight PI3K as a key driver of immunosuppression in the TME, with downstream effectors such as Akt and mTOR critically influencing the metabolic and transcriptional programs of Tregs, MDSCs, and other immunosuppressive cells [14].

Another crucial aspect of vaccine-induced immunity is antigen presentation, which involves PI3K signaling in both conventional and cross-presentation pathways. Dendritic cells (DCs), as professional antigen-presenting cells, utilize PI3K signaling to regulate antigen uptake, processing, and presentation. Specifically, PI3K activation upregulates costimulatory molecules such as CD80 and CD86, which are essential for delivering the “second signal” required for T cell activation. Additionally, PI3K signaling facilitates intracellular antigen transport and processing, promoting antigen presentation on MHC molecules [15]. In cross-presentation, PI3K signaling directs exogenous antigens to endosomal compartments through the PI3K-Akt pathway, enhancing endosomal maturation and the formation of stable peptide–MHC class I complexes [16]. PI3K inhibitors targeting isoforms such as p110δ can improve DC cross-presentation by reprogramming their maturation state, leading to more efficient CD8+ T cell activation [17].

The PI3K signaling pathway exhibits cell type-specific and isoform-dependent regulatory roles in antitumor immunity. In Tregs, PI3Kδ activation stabilizes Foxp3 expression and enhances IL-10 production while driving glycolysis and lipid metabolism to sustain survival under nutrient deprivation. The pharmacological inhibition of PI3Kδ disrupts these processes, reducing Treg abundance and restoring CD8+ T cell-mediated antitumor responses [18]. In MDSCs, PI3Kγ promotes oxidative phosphorylation and fatty acid oxidation via Akt/mTOR/STAT3 signaling, facilitating the secretion of arginase and ROS. Targeting PI3Kγ reprograms MDSCs toward a proinflammatory phenotype, enhancing T cell infiltration [19,20,21]. Conversely, balanced PI3Kδ/γ activation in cytotoxic T lymphocytes and natural killer cells supports effector functions through Akt-mediated metabolic reprogramming, including glycolysis and mitochondrial respiration. However, chronic PI3K hyperactivation may induce exhaustion markers, highlighting the need for temporally controlled therapeutic strategies [22,23].

T cell activation and proliferation rely heavily on PI3K signaling. Upon antigen recognition, Akt phosphorylation enhances cell survival, metabolic reprogramming, and clonal expansion into cytotoxic T lymphocytes (CTLs). Through mTOR, PI3K stimulates glycolysis and nutrient uptake essential for T cell proliferation [14,24]. Selective PI3K inhibition alters T cell metabolism, promoting memory T cell development [25]. Akt inhibition shifts metabolism toward oxidative phosphorylation, favoring memory T cell generation [26,27]. This finding suggests that the modulation of PI3K signaling can optimize vaccine-induced T cell responses by balancing effector function and long-term persistence (Figure 1). The PI3K pathway also regulates memory B cell and plasma cell differentiation during germinal center reactions, supporting class-switch recombination and high-affinity B cell selection [28]. PI3K signaling plays an essential role in class-switch recombination and the selection of high-affinity B cell clones, both of which are necessary for generating strong humoral immune responses (Figure 2).

The pharmacological inhibition of PI3K demonstrates antitumor efficacy through direct tumor suppression and immune modulation. PI3K blockade reduces PIP3 levels, attenuating AKT/mTOR activation to impair tumor proliferation and metastasis. It restores apoptotic sensitivity via Bcl-2 regulation and inhibits epithelial–mesenchymal transition (EMT) [29,30]. Immunomodulatory effects include reducing Tregs and MDSCs while preserving CTLs and natural killer (NK) cells. PI3K inhibition also downregulates VEGF expression and endothelial migration, impairing tumor vascularization while limiting glycolytic flux to exacerbate tumor energy stress [31,32].

The p110γ and p110δ isoforms differentially influence immune cells. p110δ, expressed in lymphocytes, drives immunosuppressive functions; its selective inhibition disrupts Treg activity and enhances antitumor immunity. p110γ, prominent in myeloid cells, regulates macrophage polarization and neutrophil functionality [33,34,35]. PI3K integrates signals from receptor tyrosine kinases (RTKs), G protein-coupled receptors (GPCRs), and cytokine receptors, activating downstream effectors like Akt and mTOR. Dysregulation disrupts immune homeostasis, contributing to tumor progression [36,37]. Clinically, the p110δ inhibitor idelalisib achieves a 72% overall response rate in hematologic malignancies [38], while AMG319 reduces Tregs in head and neck squamous cell carcinoma [39]. The p110γ inhibitor eganelisib shows modest efficacy in PD-1/PD-L1 inhibitor-resistant tumors [40] (Figure 3). Copanlisib combined with rituximab extends median progression-free survival in relapsed NHL (21.5 vs. 13.8 months, HR = 0.52) [41], suggesting enhanced TME remodeling and memory T cell responses [42].

These findings underscore the potential of PI3K-targeted therapies to synergize with cancer vaccines by enhancing immunogenicity and maintaining prolonged immune protection against tumors.

## 3. PI3K Inhibitors as a Monotherapy

PI3K inhibitors exert antitumor activity through multifaceted mechanisms targeting the hyperactivated PI3K/Akt/mTOR axis, a hallmark of numerous malignancies [43]. By suppressing PI3K-mediated PIP3 generation, these agents attenuate downstream Akt/mTOR signaling, thereby disrupting tumor cell proliferation, survival, and metabolic reprogramming. Key mechanisms include G1/S cell cycle arrest via cyclin D1 downregulation, restoration of apoptosis through BAD dephosphorylation and caspase activation, and the inhibition of glycolysis and glutaminolysis critical for tumor bioenergetics. For instance, idelalisib induces apoptosis in chronic lymphocytic leukemia (CLL) and follicular lymphoma (FL) by destabilizing Mcl-1 and Bcl-xL, highlighting its pro-death efficacy in hematologic malignancies.

Current PI3K inhibitors are categorized into pan-class I inhibitors and isoform-selective agents. Pan-PI3K inhibitors (e.g., copanlisib) broadly target α, β, γ, and δ isoforms, achieving pathway suppression but often incurring dose-limiting toxicities such as hyperglycemia and hepatotoxicity due to ubiquitous PI3K expression in normal tissues. In contrast, isoform-specific inhibitors leverage oncogenic dependencies in particular cancers: α-selective alpelisib demonstrates efficacy in PIK3CA-mutant HR+/HER2− breast cancer by selectively blocking mutant p110α, while δ-selective idelalisib exploits B cell receptor pathway reliance in lymphoid malignancies. Emerging γ/δ dual inhibitors (e.g., umbralisib) further refine specificity, mitigating immunosuppressive myeloid cell functions in marginal zone lymphoma.

The clinical validation of PI3K inhibitors reveals both promise and limitations. Approved agents like alpelisib, combined with fulvestrant, prolong progression-free survival (median 11 vs. 5.7 months) in PIK3CA-mutant breast cancer, while idelalisib–rituximab combinations achieve 83% overall response rates in relapsed CLL. However, adaptive resistance via RTK/MEK reactivation and PI3K-independent mTORC2 signaling remains a persistent challenge. Toxicity profiles also vary by specificity: pan-inhibitors like buparlisib exhibit higher rates of mood disorders and transaminitis, whereas isoform-selective agents reduce off-target effects but face immune-related adverse events (e.g., colitis with δ inhibitors) [44].

Optimizing PI3K inhibition requires biomarker-driven strategies to address heterogeneity in pathway activation and resistance mechanisms. Ongoing trials explore next-generation agents with enhanced isoform specificity (e.g., α/δ-sparing inhibitors) and combinatorial regimens pairing PI3K inhibitors with CDK4/6 inhibitors or PD-1 blockers to overcome microenvironment-mediated resistance [45,46]. Additionally, nanoparticle delivery systems and intermittent dosing protocols aim to improve therapeutic indices. Despite current challenges, the integration of genomic profiling and dual-pathway targeting positions PI3K inhibition as a cornerstone of precision oncology, warranting continued innovation to maximize efficacy while minimizing systemic toxicity [46].

Clinically, PI3K inhibitors synergize with conventional therapies to overcome drug resistance. By suppressing MDR1/P-gp efflux pumps and reversing chemotherapy-induced survival pathways, these agents enhance the cytotoxicity of platinum-based regimens and antimetabolites. Combinatorial approaches with mTOR or AKT inhibitors demonstrate additive effects, while integration with immune checkpoint blockade potentiates T cell-mediated tumor clearance. Emerging clinical data highlight improved progression-free survival in breast, lung, and colorectal malignancies, with agents like idelalisib and copanlisib demonstrating durable responses. 

The toxicity profiles of PI3K inhibitors demonstrate both class-wide commonalities and isoform-specific distinctions [47]. Pan-PI3K inhibitors such as copanlisib exhibit broad-spectrum adverse events (AEs), including hyperglycemia (5% grade ≥ 3), leukopenia, and infection-related complications such as pneumonia (8% incidence), likely attributable to ubiquitous pathway involvement in metabolic and immune homeostasis [48]. Isoform-selective agents display divergent AE patterns: α-specific inhibitors like alpelisib frequently induce hyperglycemia (60% all-grade) and mucocutaneous toxicities (rash, stomatitis), compounded by ESR1 mutation-driven resistance in hormonal cancers. δ-selective inhibitors (idelalisib, duvelisib) manifest immune-mediated toxicities, with idelalisib-treated SLL/FL patients experiencing grade ≥ 3 neutropenia (25%), hepatotoxicity (ALT elevation: 18%), and colitis (14%), while duvelisib shows higher rates of inflammatory complications (colitis: 12%). γ/θ-targeting agents (buparlisib, taselisib) are associated with hypertension (20–30%) and hepatotoxicity, whereas β-selective AZD6482 primarily induces gastrointestinal disturbances (diarrhea: 40%) and dermatologic reactions [49]. Despite tumor-type variability—notably heightened hyperglycemia in breast cancer and infection risks in hematologic malignancies—class-common toxicities (hyperglycemia, diarrhea, transaminitis, cytopenia) underscore the necessity for rigorous monitoring and prophylactic management. Therapeutic optimization requires isoform-selective targeting to mitigate toxicity while exploiting pathway dependencies in specific cancer subtypes. Future directions emphasize the development of dual PI3K/mTOR inhibitors, nanoparticle-based delivery systems, and biomarker-driven stratification to maximize therapeutic indices. Collectively, PI3K inhibition represents a paradigm-shifting strategy in precision oncology, offering a versatile platform for multimodal cancer therapy.

## 4. PI3K Inhibitors and Cancer Vaccines: A Promising Combination Strategy to Enhance Antitumor Immunity

To fully harness the therapeutic potential of PI3K-targeted strategies, the incorporation of molecular profiling and immunogenomic analyses into clinical trials is crucial. These studies play a fundamental role in validating personalized treatment approaches, optimizing patient selection, and determining the most effective timing and dosage for combination therapies to enhance therapeutic efficacy. Beyond improving cancer vaccine efficacy, PI3K inhibitors have demonstrated potential when combined with other immunotherapies, including immune checkpoint inhibitors. Preclinical studies have shown that combining PI3K inhibitors with anti-PD-1 antibodies yields synergistic effects, leading to improved tumor control [22]. These findings indicate that integrating cancer vaccines into triple-combination regimens—comprising PI3K inhibitors, checkpoint inhibitors, and vaccines—could further enhance antitumor immunity by targeting multiple immunoregulatory pathways. Ongoing early-stage clinical trials, such as the phase I/II study investigating copanlisib in combination with nivolumab (anti-PD-L1) for the treatment of microsatellite-stable (MSS) colorectal cancer (NCT03711058), are actively evaluating these synergistic approaches.

Preclinical models consistently demonstrate that PI3K inhibition can restore compromised DC function within the TME, thereby enhancing vaccine-induced immune priming [50,51]. DCs are central to initiating and sustaining antitumor immune responses by presenting tumor antigens to T cells, a process critical for activating the immune system against tumor cells. However, in the immunosuppressive TME, DC functionality is often impaired, leading to inefficient T cell activation. PI3K inhibitors have been shown to improve DC-mediated antigen presentation, thereby strengthening tumor-specific T cell responses. These mechanistic insights provide strong justification for integrating PI3K inhibitors with cancer vaccines, aiming to enhance vaccine efficacy and sustain long-term immune protection [52].

PI3K inhibition has been shown to mitigate the immunosuppressive effects of Tregs and MDSCs, both of which are critical mediators of immune evasion within the tumor microenvironment. By reducing the activity of these immunosuppressive cells, PI3K inhibitors help alleviate the suppression of effector T cells, thereby enhancing vaccine-induced T cell responses [46]. Selective p110δ inhibitors, such as idelalisib, have demonstrated the ability to restore effector T cell activity in immunosuppressive settings, underscoring their potential as adjuvants for cancer vaccines [53]. Furthermore, preclinical studies have provided evidence that combining PI3K inhibitors with cancer vaccines produces synergistic effects. PI3K inhibition has been found to enhance antigen presentation by DCs and promote the expansion of tumor-specific T cells, contributing to a more robust immune response [54] (Figure 4).

The combination of PI3K inhibitors with cancer vaccines represents a promising strategy to address the limitations associated with single-agent therapies. This approach can help overcome resistance mechanisms, enhance tumor immunogenicity, and optimize immune responses. Tumors frequently evade immune detection through mechanisms that suppress antigen presentation and inhibit T cell activation. By targeting these pathways, PI3K inhibitors contribute to the reprogramming of the tumor microenvironment, fostering stronger T cell responses and improving vaccine efficacy [24].

Recent preclinical and early clinical studies provide strong evidence supporting the synergistic effects of PI3K inhibitors and cancer vaccines. In preclinical models, PI3K inhibitors have been shown to remodel the tumor microenvironment, thereby enhancing the efficacy of cancer vaccines. The combination of PI3K inhibitors with personalized vaccines targeting tumor-specific neoantigens represents a promising therapeutic strategy. PI3K inhibition-induced immunomodulation can improve vaccine efficacy by enhancing antigen presentation and promoting T cell activation. Notably, studies have demonstrated that PIK3CA mutations alter chromatin accessibility and transcriptional programs, thereby influencing immune signaling and metabolic pathways [55]. These findings highlight the significance of integrating molecularly targeted therapies with immunotherapies to achieve enhanced therapeutic effects.

Building on these insights, future research should prioritize the development of clinical trials incorporating PI3K inhibitors with cancer vaccines and other immunotherapies, such as checkpoint inhibitors. Key areas of investigation include identifying predictive biomarkers for patient stratification, optimizing dosing regimens to balance efficacy and toxicity, and elucidating the mechanisms underlying therapeutic synergy. Ongoing trials, including the copanlisib and nivolumab combination study in MSS colorectal cancer (NCT03711058), will provide valuable clinical data on the feasibility and effectiveness of these combination strategies.

## 5. Improving PI3K Inhibitors for Combination with Cancer Vaccines

A crucial consideration in vaccine design for combination therapies is the selection of antigens and delivery platforms that optimize immune activation. Tumor neoantigens derived from patient-specific mutations serve as ideal candidates due to their high immunogenicity and tumor specificity. Delivery platforms such as mRNA vaccines and viral vectors further enhance antigen presentation and T cell priming. When used alongside PI3K inhibitors, these platforms can elicit strong immune responses while counteracting the immunosuppressive effects of the tumor microenvironment (TME). Additionally, incorporating adjuvants that stimulate innate immunity, such as Toll-like receptor (TLR) agonists, can complement PI3K inhibitors by promoting DC activation and amplifying vaccine-induced immune responses [56,57] (Figure 5).

The advancement of cancer vaccine strategies targeting PI3K relies on the development of next-generation PI3K modulators and innovative delivery methods aimed at improving therapeutic precision and efficacy. Emerging PI3K inhibitors are designed for greater selectivity toward specific isoforms, such as p110δ and p110γ, which play key roles in immune regulation [58]. An emerging research direction involves dual-function PI3K inhibitors, which selectively target both tumor cells and immunosuppressive immune cells, including Tregs and MDSCs [59]. These inhibitors have the potential to suppress tumor growth while simultaneously reprogramming the TME, thereby enhancing the effectiveness of cancer vaccines. For instance, inhibitors that selectively modulate p110δ activity in Tregs while preserving effector T cell function can potentiate vaccine-induced antitumor immunity while minimizing systemic immunosuppression [60].

Innovative delivery systems, including nanoparticles, liposomes, and micelles, are advancing therapeutic strategies in this field. These systems facilitate the co-delivery of PI3K inhibitors and vaccine components, enabling synchronized release and targeted accumulation within the TME or immune cells. For example, nanoparticles encapsulating PI3K inhibitors and tumor antigens enhance antigen presentation while concurrently reducing MDSC and Treg activity within the TME [61]. Stimuli-responsive delivery platforms, which react to specific conditions such as the acidic pH or elevated ROS levels in the tumor environment, are under active investigation [62]. The cutting-edge technology of PI3K inhibitors promotes the precision and efficiency of tumor treatment through multidimensional innovation: dual target inhibitors (such as MTX-531) break through the limitations of single targets, synergistically inhibit the PI3K and EGFR signaling pathways, and double block the key driving factors of tumor proliferation and survival, significantly enhancing antitumor activity and reducing the risk of drug resistance [61]. Meghna et al.’s research shows that nanotechnology interacts with PI3K inhibitors (such as PIK-75) through mechanisms such as targeted delivery, improved drug solubility, and enhanced cell uptake, thereby significantly enhancing its efficacy in tumor treatment. The folate-targeted PIK-75 nanosuspension not only increased drug accumulation in tumor tissues, but also improved its bioavailability, enhanced cytotoxicity, and effectively downregulated pAkt expression in the PI3K/AKT signaling pathway [63]. Aviram et al. used P-selectin targeted nanoparticles to encapsulate the PI3K α inhibitor BYL719, achieving tumor microenvironment-specific accumulation. In a head and neck squamous cell carcinoma model, only 1/7 of the oral dose of BYL719 was needed to inhibit tumor growth and enhance radiotherapy sensitization [61]. The targeting of nanocarriers not only enhances the bioavailability of drugs at the lesion site but also effectively avoids acute and chronic metabolic toxicity (such as hyperglycemia) caused by extensive inhibition of the PI3K pathway in traditional treatments by limiting system exposure. This strategy confirms the crucial role played by nano delivery systems in expanding the therapeutic window of PI3K inhibitors, providing a new pathway to overcome dose-limiting side effects. These results indicate that nanotechnology provides new opportunities for targeted delivery and the improved therapeutic efficacy of PI3K inhibitors. Computational-driven drug design is based on X-ray crystal structure and molecular dynamics simulation, accurately optimizing drug molecular configuration, improving selectivity for PI3K subtypes and pharmacokinetic properties (such as plasma half-life and tissue distribution), thereby reducing off-target effects. The toxicity management strategy involves mechanism analysis and structural optimization (such as MTX-531 avoiding the hyperglycemic toxicity of traditional PI3K inhibitors in preclinical models), combined with targeted delivery systems (such as nanocarriers) to further reduce systemic side effects [64]. The clinical translation direction focuses on expanding indications and developing combination therapies while utilizing biomarker screening to achieve personalized treatment. The integration of these technologies marks a new era of PI3K inhibitors moving from single-pathway inhibition to precise, safe, and synergistic therapy. These systems enable controlled payload release, thereby minimizing systemic toxicity and enhancing the therapeutic index of combination treatments (Figure 6).

The administration time of PI3K drugs depends on their mechanism of action and target. PI3K I plays a crucial role in acute and early neutrophil migration; therefore, for early inflammatory responses, PI3K inhibitors should be used as early as possible after administration, such as within the first 90 min after MIP-2 or KC treatment. As the inflammatory response persists, the effect of PI3K I gradually weakens, while PI3K II takes effect at later time points (such as 4 to 5 h later). At this point, PI3K II inhibitors can effectively inhibit the migration of neutrophils. Therefore, according to the time course of inflammation, the use of PI3K inhibitors should be initiated in the early stages of inflammation with PI3K I inhibitors, and in the later stages with PI3K II inhibitors.

Additionally, artificial intelligence (AI) is accelerating the discovery and optimization of PI3K inhibitors and delivery systems. AI-driven algorithms facilitate the prediction of drug–tumor interactions, optimization of inhibitor structures for isoform specificity, and design of delivery platforms tailored to individual patient profiles [65]. For instance, AI can aid in identifying tumor molecular characteristics, predicting sensitivity to PI3K inhibition, and contributing to the development of highly targeted therapeutic approaches [66].

The utilization of biomarkers for patient stratification may play a crucial role in optimizing PI3K-targeted cancer vaccine strategies. Genetic alterations such as PIK3CA mutations, PTEN loss, or dysregulated mTOR signaling can identify patients with hyperactivated PI3K pathways, who are more likely to benefit from targeted therapies [67,68]. Patients with PIK3CA-mutant tumors may demonstrate improved responses to PI3Kα-specific inhibitors, such as alpelisib, when used in combination with cancer vaccines [69,70].

CRISPR-based gene editing is emerging as a valuable tool for precision medicine. By precisely modifying components of the PI3K pathway, CRISPR technology has the potential to enhance PI3K-targeting strategies and improve vaccine efficacy [71]. For instance, the CRISPR-mediated knockout of inhibitory PI3K regulators in dendritic cells can enhance antigen presentation and cross-priming of cytotoxic T cells. Additionally, CRISPR-based screening can identify novel regulators of the PI3K pathway, providing new molecular targets to enhance vaccine-induced immune responses [72].

The application of multi-omics approaches, including genomics, transcriptomics, and proteomics, can further refine patient selection by offering a comprehensive assessment of PI3K pathway activation and immune landscape profiles [73,74]. Single-cell RNA sequencing, for example, can provide insights into the effects of PI3K inhibition on individual immune cell subsets, aiding in the design of combination therapies tailored to specific TME characteristics. Moreover, biomarker integration can inform dynamic dosing strategies, ensuring that PI3K inhibitors are administered at optimal time points to enhance vaccine-induced immune activation while avoiding interference with the initial inflammatory response.

## 6. Clinical Translation and Challenges

While PI3K-targeted vaccine strategies present considerable therapeutic potential, several challenges must be addressed before clinical translation. A primary concern is the management of toxicities associated with PI3K inhibitors, including hyperglycemia, hepatotoxicity, and immune-related adverse events (irAEs) [75]. Although isoform-selective inhibitors have reduced off-target effects, their immunosuppressive activity may still negatively impact vaccine-induced T cell responses. Table 1 and Table 2 comprehensively outline the current landscape of PI3K-targeted research: Table 1 categorizes PI3K inhibitors and their associated clinical trials, while Table 2 systematically details preclinical investigations stratified by tumor type, collectively providing a roadmap for translational advancements in isoform-specific PI3K modulation. Early pan-PI3K inhibitors, such as buparlisib, were linked to severe hepatotoxicity, gastrointestinal inflammation, and systemic immunosuppression, which restricted their therapeutic applicability [76,77]. Despite improvements in selectivity, adverse effects remain a concern. For instance, idelalisib, a p110δ-selective inhibitor, has been associated with immune-related complications, including opportunistic infections such as Pneumocystis jirovecii pneumonia and cytomegalovirus (CMV) reactivation. This occurs due to its inhibition of p110δ, which plays a critical role in the normal function of T and B cells [60]. Similarly, alpelisib, though effective against PIK3CA-mutant cancers, frequently induces hyperglycemia by disrupting insulin signaling and glucose metabolism, often necessitating dose modifications or treatment discontinuation in some patients [70].

When PI3K inhibitors are combined with cancer vaccines and other immunotherapies, associated toxicities present significant challenges. Christopher et al.’s research shows that when used as monotherapy, the most common treatment-related AEs were hepatic transaminase elevations and skin rashes, which were dose-dependent. At the 60 mg dose, a relatively high proportion of patients experienced hepatic AEs, and no grade ≥ 3 hepatic events occurred at lower doses. Similar patterns were seen for skin AEs. In combination with nivolumab, the safety profile was consistent with that of monotherapy and nivolumab’s known AE profile, with hepatic and skin-related AEs being common. The toxicity of eganelisib could affect patients’ tolerance and compliance with treatment, leading to dose reductions or treatment discontinuation, which might indirectly influence the antitumor response [64]. The effectiveness of cancer vaccines relies on strong immune activation; however, the immunosuppressive effects of PI3K inhibitors, particularly those targeting p110δ, may interfere with vaccine priming. Additionally, hepatotoxicity and gastrointestinal inflammation have the potential to intensify the systemic inflammatory responses triggered by vaccines [78]. Furthermore, the compensatory activation of alternative pathways, such as MAPK or mTOR, frequently leads to treatment resistance, limiting the long-term effectiveness of PI3K inhibitors as monotherapies [17]. These factors highlight the necessity of combination strategies that concurrently target multiple oncogenic pathways while minimizing off-target effects.

The key to managing the toxicity of PI3K inhibitors in vaccine combinations is to adopt proactive intervention measures and comprehensive support strategies. Firstly, patients should receive comprehensive education on possible toxicity and undergo frequent monitoring in order to identify and manage adverse reactions early. Common toxicities include high blood sugar, rash, diarrhea, and stomatitis. For hyperglycemia, it is recommended to monitor blood sugar at the beginning of treatment and manage it with medications such as metformin as needed. The rash is treated according to the degree of surface area involvement and itching. Mild cases can be treated locally with potent steroids, and in severe cases, medication can be interrupted and combined with systemic steroids, etc. [70,79,80]. Diarrhea should be ruled out for other reasons first, and dietary adjustments should be made. If it persists, antidiarrheal drugs such as loperamide should be used, and in severe cases, medication should be discontinued [70,81]. If pneumonia is highly suspected, it is suggested to discontinue medication, evaluate other causes, and after diagnosis, use high-dose systemic corticosteroids and discontinue PI3K inhibitors. For severe toxic reactions, reducing drug dosage or temporarily interrupting treatment should be considered. Most of the toxicity is reversible due to the short half-life of the drug, and early intervention can control it.

Several strategies are currently under investigation to address these challenges. The development of next-generation PI3K inhibitors with improved selectivity and optimized dosing regimens may help reduce toxicity while preserving therapeutic efficacy. For instance, intermittent dosing schedules for drugs such as idelalisib have demonstrated the ability to mitigate hepatotoxicity without compromising antitumor activity. Additionally, combinatorial approaches involving PI3K inhibitors with immune checkpoint inhibitors or corticosteroids are being explored to manage immune-related adverse events effectively. Biomarker-driven strategies could further assist in identifying patient subgroups most likely to benefit from PI3K inhibition while reducing the risk of severe toxicity. These approaches offer the potential to overcome the current limitations associated with PI3K inhibitors, thereby expanding their applicability in combination with cancer vaccines and other immunotherapeutic interventions.

Clinical trials have demonstrated that PI3K inhibitors can restore immune function and overcome resistance in extensively pretreated patients. In the CHRONOS-3 trial, the combination of copanlisib, a pan-class I PI3K inhibitor, with rituximab significantly improved the ORR (81% vs. 48%) and median PFS (21.5 months vs. 13.8 months) in patients with relapsed indolent non-Hodgkin lymphoma (iNHL) compared to rituximab alone [41]. Similarly, in the phase II DYNAMO trial, duvelisib, an orally administered dual PI3K-δ/γ inhibitor, achieved an ORR of 47% in patients with refractory iNHL, with a median response duration (DOR) of 10 months and a median PFS of 9.5 months [82]. These findings underscore the capacity of PI3K inhibitors to reinstate immune activity and counteract resistance in heavily pretreated patients.

Insights from animal models further emphasize the importance of timing in combination therapies. In a mouse melanoma model (B16F10-OVA), the simultaneous administration of a PI3Kδ inhibitor (PI-3065) with vaccination suppressed key inflammatory signals, such as IL-12 and TNF-α, which are critical for DC activation and effector T cell recruitment. This disruption impaired immune responses and compromised tumor control [83,84]. To address this issue, delaying PI3K inhibitor administration by 24–48 h after vaccination has been suggested. This approach preserves the vaccine-induced immune priming phase, allowing the inflammatory cascade to proceed unimpeded. During the subsequent effector phase, when Tregs and MDSCs begin exerting suppressive effects within the TME, PI3K inhibition can be more effectively utilized. In the same mouse model, delayed administration enhanced CD8+ T cell responses and improved tumor control compared to simultaneous treatment [85].

The effectiveness of combination therapies relies on the ability to tailor treatment regimens based on individual patient profiles. Molecular and immunological characteristics, such as PIK3CA mutations or amplifications, play a critical role in identifying patients most likely to benefit from PI3K inhibitors, as these alterations are strongly associated with hyperactivation of the PI3K pathway. Mutations in PIK3CA contribute to immune reprogramming by increasing the production of proinflammatory cytokines, including IL-6 and IL-8, which are key factors in TME remodeling. These cytokines not only facilitate tumor progression through enhanced angiogenesis and immunosuppression but also establish a pro-tumor inflammatory environment. PIK3CA inhibitors, such as alpelisib, have demonstrated the ability to suppress tumor growth by selectively targeting the overactive PI3K pathway and reducing IL-6 and IL-8 secretion. This dual effect not only inhibits tumor cell proliferation but also modulates the immune microenvironment, potentially reducing immunosuppression and creating a more favorable setting for vaccine-induced antitumor immune responses.

## 7. Conclusions

The PI3K signaling pathway represents a crucial target for enhancing the efficacy of cancer vaccines, as it simultaneously regulates tumor progression and immune suppression within the TME. The development of next-generation isoform-selective inhibitors, advanced delivery systems, and emerging technologies such as AI and CRISPR presents transformative opportunities to optimize these therapeutic strategies, allowing for the precise modulation of immune responses while reducing toxicity. The integration of PI3K inhibitors with innovative vaccine platforms and complementary therapies has the potential to overcome existing limitations and maximize the effectiveness of immunotherapy. Continued research, clinical validation, and interdisciplinary collaboration will be essential in translating these advancements into durable and effective treatments, ultimately contributing to improved patient outcomes.

## Figures and Tables

**Figure 1 vaccines-13-00292-f001:**
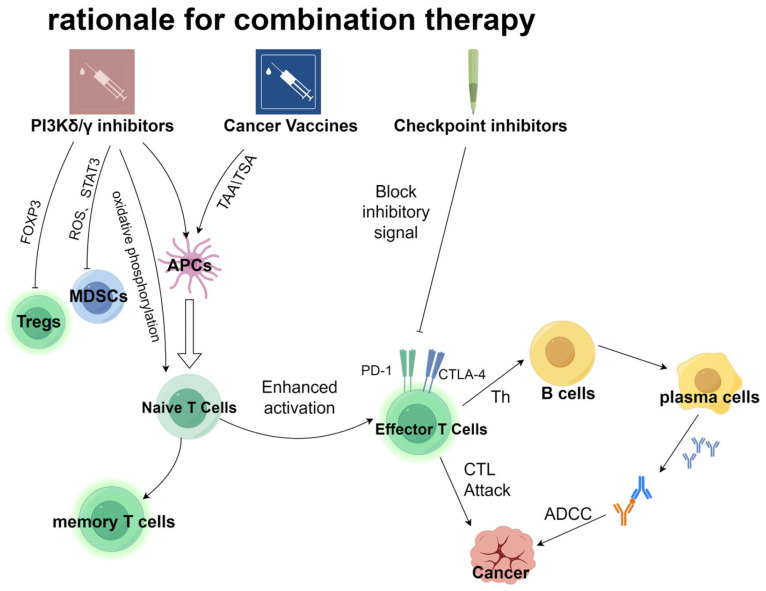
PI3K signaling pathway inhibition and antitumor immune synergistic therapy strategy. Inhibiting the PI3K signaling pathway can enhance antitumor immune response through dual regulation of immune activation and immune suppression, on the one hand activating effector immune mechanisms, and on the other hand inhibiting the function of immune-suppressive cells such as regulatory T cells (Tregs) and myeloid-derived suppressor cells (MDSCs). On this basis, tumor vaccines using tumor-associated antigens (TAAs) or tumor-specific antigens (TSAs) can enhance immune response function, while immune checkpoint inhibitors further alleviate immune suppression by blocking inhibitory signals. The synergistic application of three types of therapies significantly improves the targeted clearance efficiency of cancer cells, providing multidimensional strategic support for antitumor immunotherapy.

**Figure 2 vaccines-13-00292-f002:**
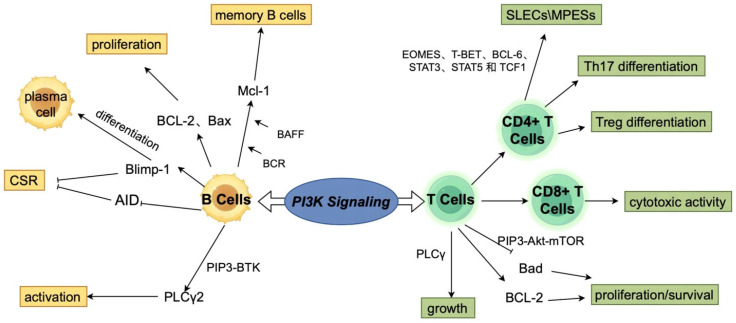
The regulatory role of PI3K signaling pathway in immune cell function. The PI3K signaling pathway widely regulates the activation, differentiation, and survival of immune cells such as T cells and B cells by integrating signals transmitted by multiple receptors. Its functions include immune cell proliferation, cytokine secretion, and antigen presentation ability, thereby coordinating the dynamic balance of innate and adaptive immune responses, providing support for the body’s immune defense and homeostasis maintenance.

**Figure 3 vaccines-13-00292-f003:**
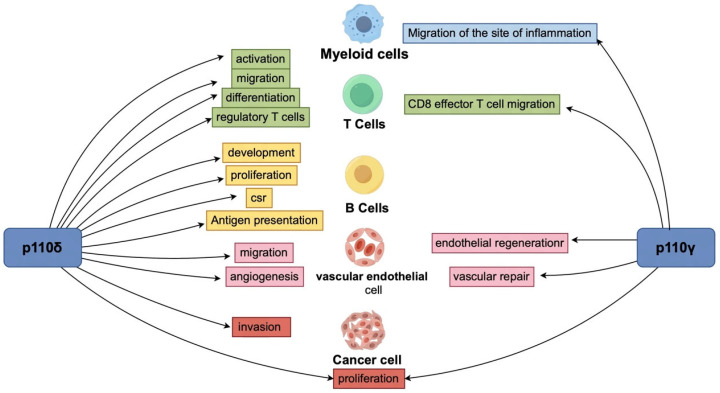
P110δ and P110γ are key subtypes of the PI3K signaling pathway, regulating multiple cellular functions in the tumor microenvironment. P110δ mainly affects the activation and proliferation of T cells and B cells, while P110γ regulates tumor inflammatory responses in myeloid cells. Together, they influence the angiogenesis of vascular endothelial cells and the growth, invasion, and metastasis of tumor cells, playing an important role in the immune regulation of TME and tumor progression.

**Figure 4 vaccines-13-00292-f004:**
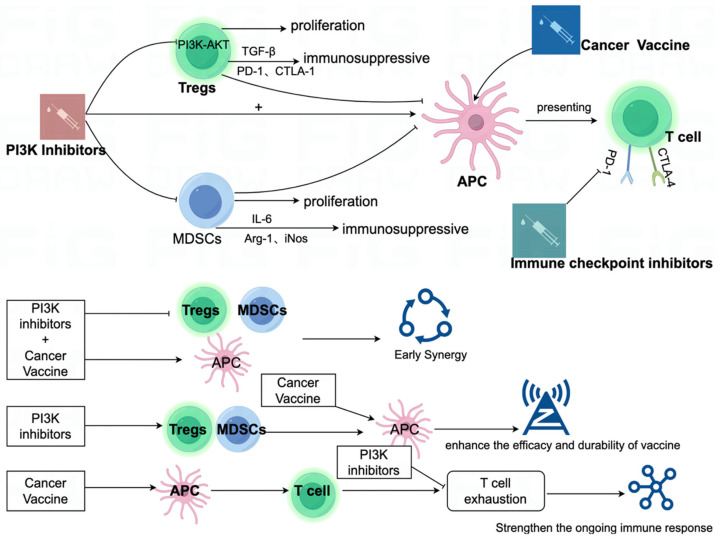
The PI3K signaling pathway profoundly influences immune responses triggered by cancer vaccines by modulating the tumor microenvironment and immune cell activities. Activation of PI3K/Akt signaling in Tregs and MDSCs promotes their proliferation and immunosuppressive functions, facilitated by factors like TGF-β, PD-1, CTLA-1, IL-6, Arg-1, and iNOS, which inhibit APCs and T cell activation. PI3K inhibitors can counteract these effects by inhibiting Tregs and MDSCs, thereby enhancing APC function and promoting T cell activation. When combined with cancer vaccines, PI3K inhibitors foster early synergy, improving antigen presentation and stimulating prolonged immune responses. Moreover, combining PI3K inhibitors with immune checkpoint inhibitors can reduce T cell depletion and boost sustained immune responses, thereby enhancing the efficacy and durability of cancer vaccines.

**Figure 5 vaccines-13-00292-f005:**
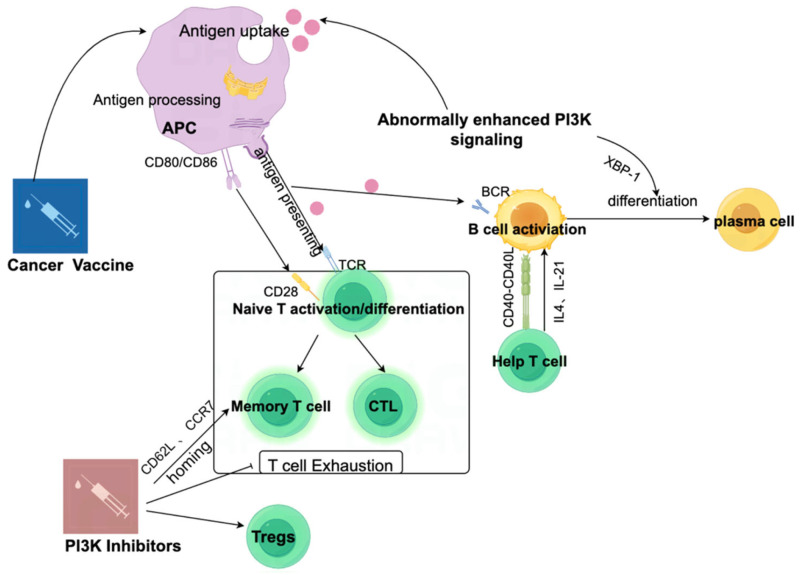
The development of PI3K inhibitors represents a promising strategy to enhance the efficacy of cancer vaccines through the modulation of immune responses. Cancer vaccines enable APCs to process and present antigens, stimulating naïve T cells through interactions with CD80/CD86 and TCRs. This leads to the differentiation of T cells into memory T cells and CTLs while simultaneously activating B cells and driving their maturation into plasma cells through the PI3K pathway. However, excessive PI3K signaling may lead to immune suppression, manifesting as T cell exhaustion and expansion of Tregs. PI3K inhibitors mitigate these effects by reducing Tregs and restoring T cell homing and activation, thereby boosting memory and effector T cell responses. When used alongside cancer vaccines, PI3K inhibitors enhance immune activation, strengthening both cellular and humoral responses to produce a more robust antitumor effect.

**Figure 6 vaccines-13-00292-f006:**
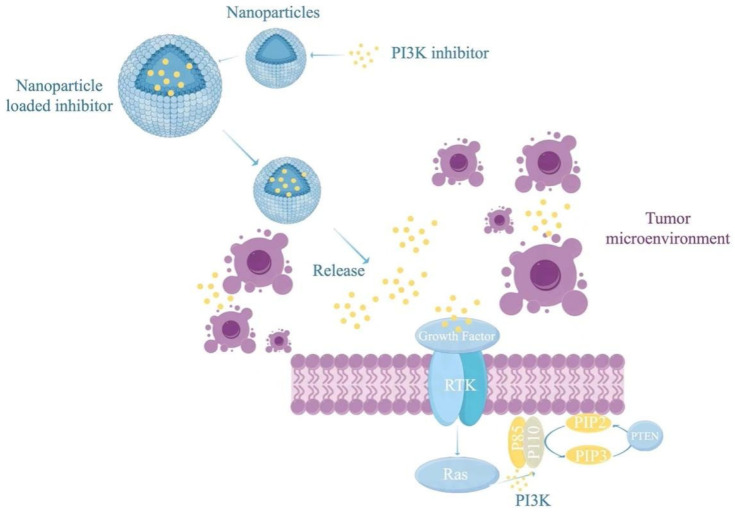
Nanoparticle or liposome encapsulation provides a promising strategy to improve the delivery and effectiveness of PI3K inhibitors and cancer vaccines within the tumor microenvironment. Encapsulation protects the PI3K inhibitors from premature degradation and ensures targeted release at the tumor site, improving drug bioavailability and reducing off-target effects. The nanoparticles are engineered to deliver their payload near tumor cells, enabling the inhibitors to effectively interfere with the PI3K signaling pathway by targeting receptor tyrosine kinases (RTKs) and downstream signaling molecules such as PIP3, thereby inhibiting tumor growth. This approach not only enhances the therapeutic efficacy of PI3K inhibitors but also provides a platform for co-delivery with cancer vaccines, synergistically modulating the immune response and improving antitumor efficacy.

**Table 1 vaccines-13-00292-t001:** Ongoing trials of PIK3CG.

NCI ID (Trial)	Phase	Type of Tumor	Type of PIK3CG	Target Enrollment	Primary Endpoint
NCT05455619	II	Breast Cancer	Evexomostat	52	AEs
NCT06530550	II	Lymphoma, NK-LGL Leukemia, T-LGL Leukemia	Linperlisib; Duvelisib	51	ORR
NCT05082025	II	Endometrial Cancer, Ovarian Cancer	copanlisib in combination with fulvestrant	7	To establish the use of copanlisib in combination with fulvestrant administered to subjects with selected estrogen receptor-positive (ER+).
NCT02164006	I	Hodgkin’s Lymphoma	TGR-1202 + brentuximab vedotin	16	ORR
NCT05306041	II	HER2-positive Breast Cancer	Inavolisib	170	ORR
NCT05676710	I	Relapsed/Refractory Large Granular T Lymphocytic Leukemia	Linperlisib	8	AEs
NCT02389842	I	Breast Cancer	Palbociclib + Taselisib/Pictilisib	79	AEs
NCT03131908	II	Melanoma	GSK2636771	27	ORR
NCT03581942	II	Primary Central Nervous System Lymphoma (PCNSL)	Copanlisib+ Ibrutinib	18	PFS
NCT01920061	I	Neoplasm	PF-05212384; Docetaxel; Cisplatin; Dacomitinib	110	AEs
NCT05387616	II	Follicular Lymphoma	Copanlisib + Obinutuzumab	98	PFS
NCT03730142	I	Advanced Cancer	WXFL10030390	82	ORR
NCT01660451	II	Lymphoma	Copanlisib	227	ORR
NCT03065062	I	Lung Cancer Squamous Cell; Solid Tumors; Head and Neck Cancer; Pancreatic Cancer	Palbociclib; Gedatolisib	96	PFS
NCT05387616	II	Follicular Lymphoma	Copanlisib + Obinutuzumab	98	PFS
NCT03711578	II	Tenalisib	Non-Hodgkin Lymphoma	20	ORR
NCT04495621	II	Metastatic Colorectal Cancer	MEN1611 + Cetuximab	29	ORR
NCT01791478	I	Breast Cancer	BYL719	46	PFS
NCT03767335	I	Breast Cancer	MEN1611 + Trastuzumab +/− Fulvestrant	62	MTD
NCT04439188	II	Lymphoma	GSK2636771	35	ORR
NCT06132932	II	PIK3CA Mutation-Related Tumors	WX390	38	ORR
NCT06224257	II	Large Granular T Lymphocytic Leukemia	Linperlisib	41	AEs
NCT01737450	II	Progressive Disease	BKM120	58	OS
NCT05683418	I	Squamous Cell Carcinoma of Head and Neck; Urothelial Carcinoma; Endometrial Cancer; HR+/HER2-negative Breast Cancer	TOS-358	241	AEs
NCT02268851	I	Lymphoma	Ibrutinib; TGR-1202	45	ORR
NCT01882803	II	Duvelisib	Indolent Non-Hodgkin Lymphoma	129	ORR
NCT04439149	II	Lymphoma	GSK2636771	35	ORR
NCT05508906	I	Breast Cancer	Drug: OP-1250 Ribociclib; Alpelisib; Everolimus	155	TEAEs
NCT06189209	II	Triple Negative Breast Cancer	Tenalisib	40	ORR
NCT04038359	II	Indolent Non-Hodgkin Lymphoma	Duvelisib	103	ORR
NCT04843098	II	Head and Neck Squamous Cell Carcinoma	TL117; Paclitaxel	108	DLT
NCT03126019	II	Lymphoma	Parsaclisib	126	PFS
NCT05021900	II	Breast Cancer	Tenalisib	40	ORR
NCT02307240	I	Triple-Negative Breast Cancer; High-grade Serous Ovarian Cancer; Solid Tumors; NUT Midline Carcinoma	CUDC-907	43	AEs
NCT03538041	II	Autoimmune Hemolytic Anemia	Parsaclisib	25	AEs
NCT04204057	II	Leukemia, Lymphocytic, Chronic, B Cell	Tenalisib	21	AEs
NCT03770000	II	T Cell Lymphoma	Tenalisib + Romidepsin	33	ORR
NCT05073250	II	Inert Non-Hodgkin’s Lymphoma	IBI376; Rituximab	40	ORR
NCT03235544	II	Lymphoma	Parsaclisib	162	ORR
NCT03586661	I	Adenocarcinoma	niraparib; copanlisib	31	Maximum tolerated dose
NCT06239467	I	Food Allergy	linvoseltamab; dupilumab	6	TEAEs
NCT03144674	II	Lymphoma	Parsaclisib	110	ORR
NCT03522298	II	Glioblastoma	Paxalisib	30	TEAEs
NCT01836861	I	Healthy	IPI-145	6	PK parameters of IPI-145 in plasma
NCT05143229	I	Breast Cancer	alpelisib + sacituzumab	18	ORR
NCT01155453	I	Advanced and Selected Solid Tumors	BKM120 + GSK1120212 DE	113	AEs
NCT05501886	III	Breast Cancer	Gedatolisib; Palbociclib; Fulvestrant; Alpelisib	701	PFS
NCT06764186	III	Breast Cancer	Capivasertib + fulvestrant	100	ORR
NCT02437318	III	Breast Cancer	Fulvestrant + alpelisib	572	OS
NCT05631795	IV	Breast Cancer	Alpelisib + fulvestrant	100	AEs
NCT02049541	I	Lymphoma	BKM120	18	ORR
NCT05768139	II	Breast Cancer; Gynecologic Cancer; HNSCC; Solid Tumors, Adult	Drug: STX-478 Fulvestrant; Ribociclib; Palbociclib	400	DLT
NCT03218826	I	Advanced Breast Carcinoma; Advanced Malignant Solid Neoplasm; Advanced Prostate Carcinoma; Anatomic Stage III Breast Cancer AJCC v8	Docetaxel; AZD8186	23	DLT
NCT03696355	I	Brain and Central Nervous System Tumors	GDC-0084	27	OS
NCT01756118	I	Leukemia	BEZ235	24	AEs
NCT04282018	II	Lymphoma	BGB-10188; Zanubrutinib; Tislelizumab	9\7	
NCT02285179	II	Breast Cancer	GDC-0032+Tamoxifen	189	AEs
NCT03688152	I	Lymphoma	INCB053914 + INCB050465	9	TEAEs
NCT02998476	II	Lymphoma	Parsaclisib	60	PFS
NCT03424122	I	B cell Lymphoma	Parsaclisib + Rituximab	50	TEAEs

TEAEs: Incidence of treatment-emergent adverse events; DLT: dose-limiting toxicity; AEs, adverse events; ORR, objective response rate; PFS, progression-free survival; OS, overall survival.

**Table 2 vaccines-13-00292-t002:** PI3KCG reported in previous studies.

Cancer Type	Cell Strain	Animal Model	Detection Index
Breast Cancer	MDA-MB-468	-	Downregulation of BRCA1/2 transcription
Melanoma	IRB 11-003254	-	ERK phosphorylation level
Breast Cancer	MCF7, ZR75-1, SUM52 and CAMA-1	-	Increase in MYC
Small Cell Lung Cancer	H69, H1048, H209	Female Balb/c nude Mice	LKB1 and pAMPK α Expression rates
Small Cell Lung Cancer	JIMT1, HCC1954	-	Tumor growth rate
Breast Cancer	BT474	-	Tumor growth rate
Lymphoma	MEC1,GM06990, JeKo-1 and Mino	Wild-type mice	AID α expression rates
Breast Cancer	BT474 and MCF7	Female mice	Average tumor volume
Glioblastoma	LN229, U87MG, GBM43, GBM5 and GBM12		Tumor growth rate
Breast Cancer	OCUB-F, SUM185PE and SUM190PT etl.	Unintentional NU/NU nude mice	Tumor growth rate
Breast Cancer	HR6	Female mice	Tumor growth rate
Breast Cancer	Met-1	FVB/N background Mice	Tumor growth rate
Prostate Cancers	Pten+/−, PtenLx/Lx and PtenLx/Lx;Trp53Lx/Lx MEFs	Mice	Apoptosis response of cells
Breast Cancer	BT-20, DLD-1, HEK293T, HCC1937, HeLa, MDA-MB-435, MDA-MB-231, NIH/3T3 and MCF7	Female nude mice	Tumor growth rate

## Data Availability

Not applicable.

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
