# Peer review of "Targeting PI3K Signaling to Overcome Tumor Immunosuppression: Synergistic Strategies to Enhance Cancer Vaccine Efficacy"

_vaccines, 2025, doi:10.3390/vaccines13030292_

Round 1
Reviewer 1 Report
Comments and Suggestions for Authors
In this review article, the authors present an overview of the phosphoinositide 3-kinases (PI3Ks), a family of intracellular lipid kinases that promote an immunosuppressive tumor microenvironment by activating regulatory T cells (Tregs), myeloid-derived suppressor cells (MDSCs) and other cell types. They also discuss the therapeutic benefits of selective PI3K inhibitors and their potential combination with immune checkpoint inhibitors (ICIs) and vaccines to improve patient outcomes.
Overall, the manuscript is focused and organized around the theme of targeting PI3K signaling to overcome tumor immunosuppression, alone and in combination with ICIs and cancer vaccines. However, almost the entire manuscript lacks quantitative information (including statistics) without which it is difficult to compare the magnitude of the effects of the PI3K signaling pathway on the biological properties of different immune cells and cancer types and the effects of its inhibition on various immunotherapy trials. The text reads as series of general conclusions and qualitative statements that really must be substantiated with hard data in the text and in figures and/or tables. Even the diagrams that are meant to help the reader understand key concepts of the article are difficult to understand because they are incomplete: for example, in Fig. 1, the cell types are not identified; in Fig. 3, the legend does not explain the biological significance of the different sequences. Additionally, the authors should indicate how their article presents an original and significant contribution to the field given similar review articles published in recent years.
Reviewer 2 Report
Comments and Suggestions for Authors
The Review by Cai and coworkers aims to describe the synergy between cancer vaccines and PI3K targeting as possible therapeutic strategy to improve the effectiveness of the former. Although it is a very interesting topic, the Review in itself is poorly structured and would need an extensive reorganization and a more in-depth discussion of some pivotal points.
MAJOR REVISIONS
First of all, description of the role of PI3K signalling in immune cells is a little bit misleading and would need more details. Indeed, based on what the authors reported, PI3K activation seems to stimulate both suppressive (Tregs and MDSCs) and anti-tumor CTLs) as well as to exert a dual effect on the same cell population (DCs). This is not unusual in immunology, but still the different effects rely on different factors (i.e., the source and/or duration of the stimulus, the presence of other molecules, the specific isoform involved, etc) that should be clarified and described. Otherwise, it is difficult to understand why PI3K inhibitors could represent an effective strategy, since it could negatively affect processes that are necessary for anti-cancer immune response, as DC maturation and T cell activation and survival. Probably, it would be useful to specify, when it is possible, the specific isoform acting in the different cell populations. Moreover, the role of PI3K in some important immune cell populations, such as NK cells and macrophages, is not discussed at all. This flaw is particularly relevant since they, particularly macrophages, are mentioned in the text and participate to the effects mediated by PI3K inhibition.
Another important flaw is the lack of details regard the preclinical studies employing PI3K inhibitors, which should be one of the main focuses of the Review. Indeed, the authors describe the evidences in a very generic manner, without details on the model used (at least the tumor type should be reported) and the specific observed effects. Indeed, is not sufficient to say that cell function is dampened or reprogrammed. How these changes have been evaluated? In terms of infiltrating cells, their functions, both? What are the molecules that are down and/or up-regulated after the challenge? Which effector functions are then improved? Killing, cytokine secretion, etc? These details are fundamental to state that PI3K inhibition could be useful in cancer therapy. Indeed, the description of the negative effects of PI3K activation can suggest that the opposite could have positive effects, but in a Review such effects must be described, not just hinted or suggested. This is particularly important for the paragraph 2, which should be the focus of the review but extends just for couple of pages. Indeed, the only detailed description of anti-tumoral PI3K effect is restricted to lines 391-402, but also the effects of PI3K as monotherapy should be discussed. In addition, considering the noticeable side effects reported for PI3K inhibitors in clinical trials, as stated by the authors in paragraph 4, it would be important to discuss whether these effects were observed also in animals and, if not, what are the differences between the approaches and which of them could be translated in clinical practice.
MINOR REVISIONS
Line 55: although the extended form is mentioned in the abstract, since this is the first time that Tregs and MDSCs appear in the main text, I would recommend the authors to use again the extended form here with the acronyms in brackets and then the acronyms throughout the text
The sentence in lines 58-60 repeats what is already stated in lines 56-58. I would recommend the authors to combine the two sentences in a single one
Lines 137-140: a reference is needed
Lines 167-174: this part does not refer to a specific isoform. Instead, it describes in general the effect of PI3K pathway on DC and macrophages. Therefore, it should be moved in the Introduction
Lines 403-415: references are needed
FIGURES
Figure 1 is poorly understandable. It is not clear whether the duale effects of the left (immune activation and Treg and MDSC inhibition) are mediated by the same or different PI3K inhibitors. It is not clear if the green cell is an activated form of the blue cell and which kind of cell it is (I guess T cell but it should be reported). It is not clear where inhibitory signals blockage and immune response support come from (PI3K targeting? Other pathways? This should be described in the main text and better depicted in the figure). Immune checkpoint inhibitors act on effector cells, not tumor cells, therefore the synergy should be better depicted. Figure caption mentions PI3K activation, but based on the main text it could also have detrimental effects
Figure 2 is scarcely informative, since the mechanisms of action in macrophages and NK cells are not described in the text. It really does not add any kind of information or graphical clarification, therefore it should be removed. On the other hand, I would recommend the authors to add more details on the pathways regulated by PI3K in NK and cells and, particularly, in macrophages, since they are mentioned in the following paragraph.
Although correct, Figure 3 does not match with the content of the paragraph in which it is put in. While the paragraph mentions the effects of the activation/inhibition of specific isoforms within TME, Figure 3 is a very general figure showing the structure of PI3K family members, therefore it should be, at most, a figure for the Introduction. However, the different class of PI3K members are not mentioned at all in the text, and a figure showing something not described in the text is not useful for the reader. Therefore, it should be removed.
REFERENCES
The reference list is mainly composed by other reviews. Although it is not a problem for general concepts (i.e. MDSCs role in tumor TME), it is poorly informative when it comes to those parts forming the core of the Review. For all the parts specifically mentioning PI3K inhibition, the authors should cite the original works, in which the reader can easily find the information of interest, without reading a second review to find them.
Reviewer 3 Report
Comments and Suggestions for Authors
The manuscript is overall OK. However, some parts of the content, particularly the figures, are overly simplified and provide limited new information for the reader. Some statements lack proper citation of the original research articles. The authors should include tables summarizing all PI3K-targeting inhibitors and ongoing clinical trials. Additionally, the resolution of Figure 3 is too low, making it unreadable. Some sections contain redundancy or repeated content, likely generated by AI, and should be manually curated. The declaration of AI usage should be placed in a dedicated declaration section rather than in the acknowledgments.
Comments on the Quality of English Languageok
Reviewer 4 Report
Comments and Suggestions for Authors
I really enjoyed reading your review! and I think a few additions could make it even stronger. Here are some thoughts:
- Clinical Challenges and Toxicity – You touch on the toxicities of PI3K inhibitors, but it might be helpful to go a bit deeper into how these could affect cancer vaccine responses. For instance, do immune-related adverse events (irAEs) from PI3K inhibitors interfere with vaccine-induced immunity? Any strategies to manage this?
- Timing of Combination Therapies – Your discussion on timing is okay, but expanding on optimal administration windows for different cancers or vaccine types could add more clarity. Any data on the best timing strategies?
- Future Directions – You mention AI and CRISPR, Maybe adding some examples of current research or clinical trials in this space would make it even more engaging. Any promising next-gen PI3K inhibitors or new delivery systems worth highlighting?
Some section-specific thoughts:
- Section 2.1 – The role of PI3K in Tregs and MDSCs is well-covered, but how does inhibition specifically influence their balance in cancer vaccine responses?
- Section 2.2 – Please include more data or references on how targeting these isoforms plays out in different cancers?
- Section 3 – Your vaccine design section is solid! Adding insights on how different platforms (mRNA, viral vectors) interact with PI3K inhibitors could be interesting.
- Section 4 – Can you give real-world examples from ongoing trials on managing PI3K inhibitor toxicity in vaccine combos? That would add practical value.
Round 2
Reviewer 2 Report
Comments and Suggestions for Authors
Revisions made by the authors improved the quality of the manuscript. However, some aspects still need to be improved:
Lines 73-138: this section should be better organized since many concepts are repeated twice. Moreover, references are needed for lines 115-138
Overall, Paragraphs 2.1 and 2.2 discuss the same things and should be merged in a single Paragraph 2 describing the effects of PI3K signaling on immune cells and TME. The short part related to PI3K inhibition (lines 195-198 and 214-217) could be moved in the following paragraph. I recommend the authors to carefully rewrite the paragraph to obtain a consistent paragraph without repetitions
There is a typo in Figure 1, effecor instead of effector
Lines 165-171: references are needed
Figure 2: as already mentioned in my previous report, a figure should not contain concepts not discussed in the text. The authors should discuss in Paragraph 2.1 also the effects of PI3K signaling on NK cells and macrophages.
Lines 194-195: the two sentences are not connected, please revise
Caption of Figure 3 does not match with Figure content, which instead depicts the effects mediated by the different isoforms
Please pay attention to Paragraphs numbering
Please add the proper references in Paragraph “PI3K Inhibitors as a monotherapy”
Lines 359-372: these parts are generic and could be moved in Paragraph 2
Lines 381-386: these sentence sound more as the conclusion of the previous paragraph (PI3K Inhibitors as a monotherapy)
Lines 373-380: data presented here are another example of combinatorial therapy. I recommend the authors to blend this part with the previous one regarding the combination of PI3K inhibitors with ICIs
I would suggest to change the title of the paragraph entitled “Key Components of Vaccine Design in Combination Therapies”. Indeed, just the first ten lines of the paragraph are dedicated to vaccines components only, while the rest of the text focuses on how to improve PI3K inhibitors. A title like “Improving PI3K inhibitors for combination with cancer vaccines” would better suit the content of the paragraph
Reviewer 3 Report
Comments and Suggestions for Authors
NA
Reviewer 4 Report
Comments and Suggestions for Authors
The manuscript can be accepted.
Round 3
Reviewer 2 Report
Comments and Suggestions for Authors
The manuscript is fine for me. The authors only have to correct "effecor" in the first figure